# Intervening with the Nitric Oxide Pathway to Alleviate Pulmonary Hypertension in Pulmonary Vein Stenosis

**DOI:** 10.3390/jcm8081204

**Published:** 2019-08-12

**Authors:** Richard W. B. van Duin, Kelly Stam, André Uitterdijk, Beatrijs Bartelds, A. H. Jan Danser, Irwin K. M. Reiss, Dirk J. Duncker, Daphne Merkus

**Affiliations:** 1Division of Experimental Cardiology, Department of Cardiology, Thoraxcenter, Erasmus MC Rotterdam, 3015 GD Rotterdam, The Netherlands; 2Department of Pediatrics/Neonatology, Erasmus MC—Sophia Children’s Hospital, 3015 GD Rotterdam, The Netherlands; 3Division of Paediatric Cardiology, Department of Paediatrics, Erasmus MC—Sophia Children’s Hospital, 3015 GD Rotterdam, The Netherlands; 4Department of Pharmacology, Erasmus MC, 3015 GD Rotterdam, The Netherlands; 5Walter–Brendel Centre of Experimental Medicine, University Hospital, LMU Munich, 80799 Munich, Germany

**Keywords:** pulmonary hypertension, nitric oxide, exercise, pulmonary vascular resistance

## Abstract

Pulmonary hypertension (PH) as a result of pulmonary vein stenosis (PVS) is extremely difficult to treat. The ideal therapy should not target the high-pressure/low-flow (HP/LF) vasculature that drains into stenotic veins, but only the high-pressure/high-flow (HP/HF) vasculature draining into unaffected pulmonary veins, reducing vascular resistance and pressure without risk of pulmonary oedema. We aimed to assess the activity of the nitric oxide (NO) pathway in PVS during the development of PH, and investigate whether interventions in the NO pathway differentially affect vasodilation in the HP/HF vs. HP/LF territories. Swine underwent pulmonary vein banding (PVB; *n* = 7) or sham surgery (*n* = 6) and were chronically instrumented to assess progression of PH. Pulmonary sensitivity to exogenous NO (sodium nitroprusside, SNP) and the contribution of endogenous NO were assessed bi-weekly. The pulmonary vasodilator response to phosphodiesterase-5 (PDE5) inhibition was assessed 12 weeks after PVB or sham surgery. After sacrifice, 12 weeks post-surgery, interventions in the NO pathway on pulmonary small arteries isolated from HP/LF and HP/HF territories were further investigated. There were no differences in the in vivo pulmonary vasodilator response to SNP and the pulmonary vasoconstrictor response to endothelial nitric oxide synthase (eNOS) inhibition up to 8 weeks after PVB as compared to the sham group. However, at 10 and 12 weeks post-PVB, the in vivo pulmonary vasodilation in response to SNP was larger in the PVB group. Similarly, the vasoconstriction to eNOS inhibition was larger in the PVB group, particularly during exercise, while pulmonary vasodilation in response to PDE5 inhibition was larger in the PVB group both at rest and during exercise. In isolated pulmonary small arteries, sensitivity to NO donor SNP was similar in PVB vs. sham groups irrespective of HP/LF and HP/HF, while sensitivity to the PDE5 inhibitor sildenafil was lower in PVB HP/HF and sensitivity to bradykinin was lower in PVB HP/LF. In conclusion, both NO availability and sensitivity were increased in the PVB group. The increased nitric oxide sensitivity was not the result of a decreased PDE5 activity, as PDE5 activity was even increased. Some vasodilators differentially effect HP/HF vs. HP/LF vasculature.

## 1. Introduction

Pulmonary hypertension (PH) is a chronic disease characterised by a mean pulmonary arterial pressure (mPAP) >25 mmHg at rest [1]. PH can have many different aetiologies, which are categorised into five groups by the WHO. The most prevalent is group II PH, in which left heart failure (LHF), valvular disease, or obstructions of the inflow/outflow tract of the left ventricle cause an upstream pressure increase in the pulmonary vasculature [1,2]. An often-overlooked cause of group II PH is partial pulmonary vein stenosis (PVS).

In PVS, one or more of the pulmonary veins are narrowed, hampering outflow of blood from the lungs into the left atrium, thereby increasing pulmonary vascular resistance. In the adult population, PVS may occur as an, albeit rare, consequence of radio-frequency ablation of atrial tissue around the pulmonary veins to treat atrial fibrillation [3]. PVS in the paediatric population is a severe congenital anomaly associated with poor outcomes [4]. Paediatric PVS can occur as a singular anomaly, but is generally associated with congenital heart defects (univentricular heart disease, ventricular septal defect, atrial septal defect or persistent arterial duct), lung disease (bronchopulmonary dysplasia (BPD)) or Down syndrome or other trisomies [5,6,7,8]. PVS is a challenging disease. Treatment of the stenosis itself, either through catheterisation or by means of surgery, often results in restenosis [7]. The resulting initially passive, isolated post-capillary PH (IpcPH) may progress to active, combined pre- and post-capillary PH (CpcPH), characterised by pre-capillary structural and functional vascular remodelling resembling pulmonary arterial hypertension (PAH; WHO classification group I PH) [9,10,11]. The use of PAH vasodilation therapy, however, can be perilous in group II PH, as vasodilation increases flow, while the outflow remains hampered, potentially resulting in an increase in capillary pressure and pulmonary oedema. Use of first-line vasodilator therapies like inhaled nitric oxide (NO), administration of NO donors or phosphodiesterase-5 (PDE5) inhibitors may therefore increase dyspnoea by causing pulmonary oedema [12]. To this day, crucial information about the effects of these therapies in PH due to PVS is lacking. Unlike LHF-induced PH, PVS-induced PH has a mixed or double phenotype. The pulmonary vasculature draining into the stenotic pulmonary veins experiences a high pressure with a relatively low flow (HP/LF), while the pulmonary vasculature draining into the remaining unaffected pulmonary veins experiences a high pressure with a relatively high flow (HP/HF) because of flow redistribution [13]. The ideal therapy would target only the HP/HF vasculature, lowering PAP with no detrimental effects associated with increased flow. To date, it is unclear whether vasodilator therapies differentially affect these two different lung regions. Therefore, we aimed to (i) assess the activity of the NO pathway in PVS-induced PH during the transition from IpcPH to CpcPH, and (ii) investigate the effect of an NO donor and a PDE5 inhibitor on the HP/HF and HP/LF phenotypes separately. To investigate this, we used a recently developed swine model of post-capillary PH by pulmonary vein banding (PVB) [14,15,16] that slowly progresses from IpcPH to CpcPH [17]. Chronic instrumentation allowed for repeated measurements of pulmonary haemodynamics in awake swine 5–12 weeks after PVB, during the progression of PH, while evaluating the activity of the NO pathway. Since exercise testing allows detection of perturbations in cardiopulmonary function that may not be apparent under resting conditions, which facilitates the assessment of disease severity [18], we performed measurements both at rest and during graded treadmill exercise.

## 2. Methods

### 2.1. Animal Experimentation

All experiments were performed conforming to the “Guiding Principles in the Care and Use of Laboratory Animals,” which are supported by the Council of the American Physiological Society, and with the approval of the Erasmus University Medical Centre Animal Care Committee (EMC3158, 109-13-09).

### 2.2. Outline of This Study

Non-restrictive inferior pulmonary vein banding (*n* = 7) or sham operation (*n* = 6) was performed on crossbred Landrace x Yorkshire swine of either sex (8 ± 2 kg). After 4 weeks, all 13 animals (23 ± 3 kg) underwent chronic instrumentation, allowing weekly haemodynamic measurements as well as determination of blood gases in awake animals. In the 8 weeks after chronic instrumentation, all animals performed biweekly exercise experiments, both under control conditions and in the presence of an endothelial nitric oxide synthase (eNOS) inhibitor. Animals also underwent two-weekly resting experiments, during which incremental doses of the NO donor sodium nitroprusside (SNP) were administered. After 12 weeks, a control exercise experiment was performed, followed by an exercise experiment after administration of the PDE5 inhibitor sildenafil. After completion of all experiments, in week 12, all 13 animals (61 ± 8 kg) were sacrificed, after which tissues were excised. Isolated small pulmonary arteries upstream of both banded (HP/LF) and unbanded (HP/HF) pulmonary veins were subsequently isolated, and responses to the NO donor and PDE5 inhibitor were assessed.

### 2.3. Pulmonary Vein Banding

Banding of the inferior venous confluent was performed on swine (*n* = 7, 9 ± 2 kg) as described by Pereda et al. [16]. In short, swine received an intramuscular (i.m.) injection of tiletamine/zolazepam (5 mg kg^−1^, Virbac, Barneveld, The Netherlands), xylazine, (2.25 mg kg^−1^, AST Pharma, Oudewater, The Netherlands) and atropine (0.5 mg) for sedation, and were subsequently anesthetised with an intravenous (i.v.) bolus of thiopental (10 mg kg^−1^ Rotexmedica, Trittau, Germany). After relaxation of the vocal chords, an endotracheal tube was inserted and secured and ventilation was started (O_2_:N_2_ (1:2)). Anaesthesia was maintained by adding isoflurane (2% vol/vol, Pharmachemie, Haarlem, The Netherlands) to the gas mixture. Prior to the banding procedure, antibiotic prophylaxis was administered (0.75 mL depomycine, 200.000 IU mL^−1^ procainebenzylpenicilline, 200 mg mL^−1^ dihydrostreptomycine, Intervet Schering-Plough, Boxmeer, The Netherlands). The skin on the right side of the animal was disinfected using iodine antiseptic solution, and the procedure was performed under sterile conditions. Through the fifth right intercostal space, the chest was opened and blunt dissection was used to expose and dissect the inferior venous confluence, which drains both inferior pulmonary lobes. Near the atrium, a surgical loop (Braun Medical Inc., Bethlehem, PA, USA) was placed around the inferior venous confluent and fixed at the resting diameter using a silk suture. To secure the ribs and close the thorax, non-absorbable USP6 braided polyester (Ø0.8 mm) was used and the skin was closed in layers using silk sutures. Administration of isoflurane was discontinued, allowing the animal to restore spontaneous ventilation, upon which the endotracheal tube was removed. Analgesia was administered by means of an i.m. injection (0.3 mg buprenorphine i.m. Indivior, Slough, UK) and a fentanyl slow-release patch (6 μg h^−1^, 48 h). Once fully awake, animals were transferred back to the animal facilities. A sham procedure was performed on six additional swine (8 ± 2 kg) as described above, where the vein was exposed and dissected free, but not banded. One PVB animal died directly after the procedure because of acute pulmonary oedema at the onset of spontaneous ventilation. Another PVB animal suffered from severe HF and was prematurely sacrificed 6 weeks after banding. The numbers of animals reported in the methods section and the data presented in the results section do not include these animals. All sham animals completed the entire study.

### 2.4. Chronic Instrumentation

The protocol for the chronic instrumentation performed has been previously described in great detail [19]. In brief, swine were sedated, anesthetised and ventilated as described above, and the chest and pericardium were opened via the fourth left intercostal space under sterile conditions. To allow haemodynamic measurements and extraction of blood samples, fluid-filled polyvinylchloride catheters (Braun Medical Inc., Bethlehem, PA, USA) were inserted into the pulmonary artery (2×), the aortic arch, the left atrium (2×) and the right ventricle. A transit-time flow probe (Transonic Systems Inc., Ithaca, NY, USA) was also placed around the ascending aorta to allow measurement of cardiac output. All catheters were tunnelled to the back and the thorax and wound were closed. Anaesthesia was terminated and analgesia was administered (buprenorphine i.m. (0.3 mg) and a fentanyl slow-release patch (12 μg h^−1^, 48 h)). Animals were allowed to recover and were transferred to the animal facilities once fully awake. Antibiotic prophylaxis was administered for seven consecutive days after the chronic instrumentation procedure in the form of 25 mg kg^−1^ amoxicillin i.v. (Centrafarm B.V. Etten-Leur, The Netherlands) and 5 mg kg^−1^ gentamycin i.v. (Eurovet, Bladel, The Netherlands).

### 2.5. Experimental Protocols

Experiments were performed at 6, 8, 10, and 12 weeks after PVB. Swine were placed on an adapted motor-driven treadmill and implanted catheters were connected to pressure transducers (Combitrans pressure transducers, Braun, Melsungen, Germany). Transducers and the transit-time flowprobe were connected to an amplifier. Cardiac output, heart rate and aortic, pulmonary arterial, right ventricular and left atrial blood pressures were continuously recorded. Haemodynamic measurements and blood samples were taken while the animal was resting quietly, after which a four-stage incremental treadmill exercise was initiated (1–4 km h^−1^, 3 min per speed). At rest and during the last minute of each exercise stage, when haemodynamics were steady, haemodynamic measurements were performed, and blood samples were collected. After the exercise experiments, animals received 60 min of rest, to allow haemodynamics to return to baseline. Thereafter, the eNOS inhibitor N^ω^-nitro-L-arginine (NLA; 20 mg kg^−1^, dissolved in saline to 5 mg mL^−1^) was administered by an infusion of 50 mg min^−1^ i.v. Ten minutes after complete NLA administration, the exercise experiment was repeated. On a different day, with a minimum of 2 days in between, swine were transferred to a small enclosure and fluid-filled catheters were connected to pressure transducers and amplifiers, as described above. While resting quietly, haemodynamics were continuously recorded. After baseline measurements, sodium nitroprusside (SNP) was continuously administered in incremental doses of 0.5, 1, 2, 3, 4 and 5 μg kg^−1^ min^−1^ through the second fluid-filled catheter implanted in the main pulmonary artery. Each dose was administered for 10 min, and haemodynamic measurements were performed in the final minutes of each dose.

Flowprobe data of one PVB animal were not available because of technical failure. Consequently, the number of animals for flow-related parameters (cardiac output, stroke volume, vascular resistance) was reduced to six. One sham animal as well as one PVB animal suffered from recurrent lameness and did not perform all exercise experiments, resulting in an available group size of five sham animals and six PVB animals (five for PVB flow-related parameters).

Twelve weeks after PVB or sham operation, swine (PVB *n* = 4, Sham *n* = 5) were transferred to the treadmill, catheters were connected and an exercise experiment under control conditions was performed as described above. After 60 min of rest, allowing haemodynamics to return to baseline, 10 mg of the PDE5 inhibitor sildenafil was administered in 10 min through the second catheter implanted in the main pulmonary artery. Ten minutes after completion of administration, the exercise protocol was repeated.

### 2.6. Blood Gas Measurements

pO_2_ (mmHg) was determined in the collected arterial blood samples using a blood gas analyser (ABL 800, Radiometer, Denmark).

### 2.7. Data Analysis

Haemodynamic data were recorded digitally and analysed offline, as described previously [20]. CO was corrected for bodyweight (cardiac index, CI). Total pulmonary vascular resistance index (tPVRi) and systemic vascular resistance index (SVRi) were calculated as the ratio of mPAP and CI and MAP and CI, respectively. Stroke volume index was calculated as CI/heart rate.

### 2.8. Sacrifice

After completion of follow-up, 12 weeks after PVB or sham surgery, animals were sacrificed. Animals were sedated, as described above, and anesthetised with pentobarbital sodium (i.v. 6–12 mg kg^−1^ h^−1^). After intubation, animals were ventilated with a mixture of O_2_ and N_2_ (1:2). Sternotomy was performed to expose the heart and lungs. Ventilation was terminated, and the heart was arrested and immediately excised together with the lungs. Parts of the upper (cranial) and lower (caudal) lobe of the right lung were placed in cold Krebs buffer for dissection of pulmonary small arteries for wire myograph experiments, and parts were snap-frozen for protein isolation to determine eNOS expression.

### 2.9. Wire-Myograph Experiments

In vitro pulmonary vessel experiments were performed as described previously [21,22]. Briefly, pulmonary small arteries were isolated and stored overnight in oxygenated (95% O_2_/5% CO_2_) Krebs bicarbonate solution (composition in mM: 118 NaCl, 4.7 KCl, 2.5 CaCl_2_, 1.2 MgSO_4_, 1.2 KH_2_PO_4_, 25 NaHCO_3_, glucose 8.3; pH 7.4) at 4 °C. The following day, segments 2 mm in length were cut and installed on wire myographs in separate organ baths filled with oxygenated Krebs bicarbonate solution at 37 °C. Following a half hour stabilisation period, internal vessel diameter was set to a tension equivalent to 0.9 times the estimated diameter at 20 mmHg effective transmural force. Vessels were exposed to 100 nM of the synthetic thromboxane analogue U46619 to induce pre-constriction, and endothelial integrity was ascertained subsequently by exposure to the endothelium-dependent vasodilator substance P (10 nM). To test maximal constriction, 100 mM KCl was administered, after which the vessels were washed and allowed to stabilise in fresh buffer solution for half an hour. After preconstriction with 100 nM U46619, vessels were subjected to one of three experimental protocols: incremental concentrations of the NO donor SNP (10^−9^–10^−5^ M); the PDE5 inhibitor sildenafil (10^−10^–10^−5^ M); or the endothelium-dependent vasodilator bradykinin (10^−10^–10^−7^ M), in the absence or presence of pre-incubation with L-N^ω^-nitroarginine methyl ester (L-NAME, 10^−4^ M), 30 min before administrating bradykinin. In each protocol, four–six vessel segments per lobe per group were used.

The designated software was used to analyse data (Labchart 8.0, AD instruments, Sydney, Australia) and to create concentration response curves (Prism 5.0, Graphpad Software, Inc., La Jolla, CA, USA).

### 2.10. eNOS Protein Expression

Snap-frozen lung samples were homogenised to determine total eNOS protein expression, as well as eNOS monomer and dimer protein expression. For eNOS monomer and dimer fraction detection, low temperature sodium dodecyl sulfate–polyacrylamide gel electrophoresis (SDS-PAGE) was performed, as described previously [23]. In brief, all gels and buffers were equilibrated at 4 °C before electrophoresis, and during electrophoresis, the buffer tank was placed in an ice bath to maintain low temperature. SDS-PAGE for total eNOS protein content and housekeeping protein α-tubulin was performed at room temperature. Subsequently, the proteins were transferred to nitrocellulose membranes and the blots were probed with primary, anti-eNOS (1:500, Transduction Laboratory) and anti-α-tubulin (1:10,000, Imgenex). All blots were analysed using the Odyssey system (LI-COR). Band intensities were determined, and the ratio of the intensity of the eNOS monomer and dimer as well as eNOS and α-tubulin were calculated and expressed as arbitrary units (A.U.). Monomer:α-tubulin and dimer:α-tubulin were calculated as monomer ÷ (monomer + dimer) × eNOS ÷ α-tubulin and dimer ÷ (monomer + dimer) × eNOS ÷ α-tubulin, respectively.

### 2.11. Statistical Analysis

Statistical analyses were performed using SPSS (version 21.0 IBM, Armonk, NY, USA) and comprised unpaired (two tailed) *t*-test, or one-way or two-way ANOVA for repeated measures, followed by post-hoc testing with Bonferroni when appropriate. Statistical analyses on concentration response curves consisted of regression analyses using Prism (version 5, GraphPad Software, La Jolla, CA, USA). Data were considered statistically significant when *p* ≤ 0.05 (two-tailed). Data are presented as means ± SEM.

## 3. Results

### 3.1. Induction of Pulmonary Hypertension

Pulmonary vein banding resulted in an increased tPVRi as early as 6 weeks after banding (Table 1, 132 ± 4 vs. 107 ± 9 mmHg L^−1^ min kg in sham (*p* ≤ 0.05)). Together with the maintained CI, this resulted in an increased mPAP (Table 1, 30 ± 2 vs. 20 ± 1 mmHg in sham (*p* ≤ 0.001)). Both tPVRi and mPAP increased over time (tPVRi week 12: 244 ± 26 vs. 116 ± 10 mmHg L^−1^ min kg in sham (*p* ≤ 0.01)), mPAP week 12: 39 ± 2 vs. 20 ± 1 mmHg in sham (*p* ≤ 0.001)), reflecting the progressive nature of PH. Pulmonary vein banding had minor effects on the systemic circulation at rest, as only MAP tended to be lower 12 weeks after banding, while LAP, HR, CI, SVi and SVRi were unaltered compared to sham (Table 1).

### 3.2. Response to Exercise

Exercise up to 4 km h^−1^ produced an increase in mPAP in sham animals that was the consequence of an increased CI and LAP and a maintained tPVRi (Table 1). The exercise-induced increase in LAP was similar in the PVB animals, although, 8 weeks after banding, LAP was lower in PVB compared to sham. From week 6 until week 10, the exercise-induced increase in CI (Table 1) was similar in PVB compared to sham animals, although the CI during exercise was lower in PVB in week 10 as compared to week 6. Furthermore, in week 12, CI was lower during exercise in PVB animals than in sham animals (Table 1, CI at 4 km h^−1^: 0.23 ± 0.01 vs. 0.28 ± 0.02 *p* ≤ 0.05), which was principally due to an attenuated exercise-induced increase in heart rate in PVB animals compared to sham animals (Table 1, heart rate at 4 km h^−1^: 207 ± 6 vs. 251 ± 10 beats min^−1^
*p* ≤ 0.05). Despite this attenuated increase in CI, the exercise-induced increase in mPAP was similar in PVB compared to sham animals, as a result of the higher tPVRi in PVB animals that was only significantly elevated further by exercise in week 6 (Table 1).

### 3.3. Nitric Oxide Production and Sensitivity

To assess the vasoactive effect of endogenous NO production, the eNOS inhibitor NLA was administered. In week 6 and 8, vasoconstriction response to NLA was similar in PVB and sham animals, reflected by the similar increase in SVRi and tPVRi both at rest and during exercise (Figure 1 and Figure 2). While in week 10 and 12, the increase in SVRi upon NLA administration was similar in PVB and sham, the increase in tPVRi was significantly more pronounced in PVB animals, especially during exercise (Figure 2), suggesting an augmented, exercise-induced increase of NO production. To investigate NO sensitivity, the NO donor SNP was administered (0.5–5 μg kg^−1^ min^−1^). While systemic sensitivity to SNP was transiently lower in PVB animals at week 6, there was no statistically significant difference in ΔSVRi, and thus NO sensitivity, at weeks 8, 10 or 12. In the pulmonary vasculature however, the NO sensitivity was significantly increased in PVB animals at week 10, and tended to be increased at week 12 (*p* = 0.09, Figure 2). Our data suggest that both endogenous NO production during exercise and NO sensitivity were increased after PVB.

### 3.4. Contribution of PDE5

To test if the increased sensitivity to NO was due to a decreased PDE5 activity, the PDE5 inhibitor sildenafil was administered at week 12. In sham animals, PDE5 inhibition resulted in a small decrease in SVRi and tPVRi at rest, as well as during exercise (Figure 3). In PVB animals, PDE5 inhibition resulted in considerable vasodilation in both the systemic and pulmonary vasculature at rest. During exercise, this effect waned in the systemic vasculature, but was unaltered in the pulmonary vasculature. This indicates that the increased NO sensitivity cannot be attributed to decreased PDE5 activity. Importantly, pulmonary vasodilation due to sildenafil was not accompanied by a further decrease in pO_2art_ (Figure 3), indicating that sildenafil did not induce pulmonary oedema.

### 3.5. Regional Differences in NO Pathway

To test the NO pathway separately in the HP/LF and HP/HF vasculature, isolated pulmonary small arteries from the HP/LF (caudal, banded) and HP/HF (cranial, unbanded) pulmonary lobes were pre-constricted with U46619, and subjected to incremental doses of SNP or sildenafil. SNP produced a similar dose-dependent vasodilation in pulmonary small arteries from HP/LF and HP/HF lobes of PVB animals, compared to pulmonary small arteries from corresponding lobes of sham animals (Figure 4). PDE5 inhibition with sildenafil produced a similar vasodilation in pulmonary small arteries between PVB HP/LF lobes and sham. However, pulmonary small arteries from PVB HP/HF lobes showed a smaller vasodilation than sham, suggesting a lower PDE5 activity in HP/HF lobes (Figure 4). Vasodilation response to bradykinin was similar in pulmonary small arteries from PVB HP/HF and sham, but was lower in PVB HP/LF as compared to sham. Prior inhibition of eNOS by L-NAME decreased the response to bradykinin to a similar extent in all vessels from PVB and sham animals (Figure 4). Thus, the difference in response to bradykinin between pulmonary small arteries from PVB HP/LF and sham persisted, suggesting that the decreased vasodilation response to bradykinin was NO-independent. 

Total eNOS expression was similar in the HP/LF (banded) lobe as compared to upper and lower lobes from sham animals, while eNOS expression was significantly higher in the HP/HF as compared to the HP/LF lobes (Figure 5). The eNOS monomer:dimer ratio was not altered by PVB, but, due to the higher expression in HP/HF, both eNOS monomer and dimer were increased as compared to the HP/LF territory.

## 4. Discussion

The main findings in this study were that (i) banding of the confluence of the lower pulmonary veins in swine resulted in PH; (ii) 10 weeks after banding, pulmonary vasodilation response to exogenous NO was higher in PVB as compared to sham in vivo; (iii) the endogenous NO-mediated vasodilator influence was significantly higher in the pulmonary vasculature of PVB as compared to sham animals and the difference increased over time, particularly during exercise, which was accompanied by a higher eNOS expression in the PVB HP/HF territory; (iv) the pulmonary vasodilation in response to PDE5 inhibition was enhanced in PVB as compared to sham animals both at rest and during exercise; (v) sensitivity of isolated pulmonary small arteries to NO donor SNP was similar in PVB vs. sham irrespective of HP/LF and HP/LF; (vi) sensitivity to PDE5 inhibitor sildenafil was similar in PVB HP/LF vs. sham but lower in PVB HP/HF; (vii) vasodilation to bradykinin was similar in PVB HP/HF and sham, but was lower in PVB HP/LF; (viii) this decreased response to bradykinin was NO-independent. The implications of our findings are discussed below.

### 4.1. Pulmonary Hypertension Associated with PVS

Although PVS is a rare phenomenon, the frequency of PH in PVS is very high, being up to 67% as reported by Mahgoub et al. [7], and right heart failure is the most common cause of death in this population. Treatment of PVS currently consists of either surgical pulmonary venoplasty, or a catheter-based approach like balloon angioplasty with or without intravascular stent placement. Unfortunately, treatment of PVS is technically challenging; restenosis is frequent and event-free survival is poor [4]. A French cohort of premature infants with PVS and PH reported a 44% mortality 1 year after diagnosis [8]. Despite the small number of enrolled patients, they were able to analyse 1-year survival in patients that received any type of intervention (69%) vs. patients that received no intervention (21%). A recently published multicentre retrospective cohort study of PVS affecting ex-premature infants [7] and a meta-analysis [4] found similar survival rates—between 55 and 70%. These studies identified a number of risk factors that result in decreased survival and increased risk of restenosis; among them were the number of stenotic pulmonary veins (three or more affected veins had poorer outcome then two or fewer stenotic pulmonary veins), bilateral disease and small body size for gestational age. Another important determinant of mortality is the severity of PH and concomitant RV dysfunction [24].

PH secondary to PVS is classified as group II PH because of the post-capillary position of the stenosis, which results in a left ventricular inflow tract obstruction. The IpcPH that is the result of hampered outflow from the lungs to the LV may ultimately affect the pulmonary vasculature and lead to CpcPH, characterised by pre-capillary structural and functional vascular remodelling resembling pulmonary arterial hypertension (PAH; WHO classification group I PH) [9,10,11]. Similarly, in our swine model, PVS led to PH and RV dysfunction [14,15,16]. At first, PVS-induced PH was a purely passive increase in pulmonary pressure and resistance (IpcPH), but within 12 weeks, structural and functional remodelling of the pulmonary vasculature introduced a pre-capillary component, producing CpcPH [14,17].

Most patients with PVS [25], as well as swine with PVS [15], show a decrease in pulmonary artery pressure upon vasoreactivity testing with inhaled 100% oxygen. In our previous study, we showed that ET_A/B_ blockade with tezosentan resulted in a decrease in tPVRi in swine with PVB, and that this vasodilator effect increased over time [17]. In the present study, the NO donor SNP and the PDE5 inhibitor sildenafil both induced pulmonary vasodilation that was larger in swine with PVB. However, the effect of SNP was relatively modest as compared to the reduction in tPVRi induced by PDE5 inhibition. Although it could be suggested that the dosage of SNP should be further increased to increase its effect, this was not possible, given its potent hypotensive effect on the systemic vasculature. Together with the observations that eNOS inhibition produced a larger increase in tPVRi in PVB as compared to sham, and the relatively modest vasodilation induced by SNP as compared to PDE5 inhibition, this suggests that the NO pathway is hyperactive in PVB pulmonary vasculature, probably as a compensatory mechanism, and that adding additional NO has a very limited effect. These data are consistent with the study of Domingo et al., which showed that NO on top of 100% oxygen did not induce further pulmonary vasodilation [25]. Interestingly, eNOS expression was higher in HP/HF as compared to HP/LF, and tended to be higher compared to the corresponding lung lobe of sham animals. This increased expression predominantly involved a higher expression of the NO-producing eNOS dimer, as the superoxide-producing eNOS monomer was higher in HP/HF as compared to HP/LF, but was similar in the corresponding lobes of sham animals.

In group II PH, extreme caution should be taken in the utilisation of vasodilation therapy. The guidelines for the diagnosis and treatment of pulmonary hypertension are very clear in their advice to treat only the underlying disease in group II PH [26], because an increase in flow may result in an increase in pulmonary capillary pressure and ensuing pulmonary oedema. It is, however, important to realise that in PVS, the pulmonary vasculature comprises two territories; those draining into stenotic pulmonary veins, and those draining into unaffected pulmonary veins. While the pulmonary vasculature in the first territory will experience high pressure and low flow, the latter experiences high pressure and high flow. Issues regarding pulmonary oedema that arise in treatment of PH secondary to left heart failure may be absent when treating only the HP/HF vasculature in PVS-induced PH.

This model is unique in its ability to allow for independent interrogation of these two subsets of arteries. We have previously shown that in this model, PVB leads to increased wall thickness, increased wall:lumen ratio and a decreased relative lumen area in pulmonary small arteries from both the HP/HF and HP/LF territories [17]. This structural remodelling was accompanied by increases in maximal constriction to KCl and U46619 in all pulmonary small arteries, irrespective of flow. Interestingly, in the present study, vasodilation in response to bradykinin was smaller in pulmonary small arteries from the PVB HP/LF territory than sham, while pulmonary small arteries from the HP/HF territory did not show a difference as compared to sham (Figure 4). These data are consistent with our previous observation that vasodilation to Substance P, a measure for endothelial function, was only decreased in PVB HP/LF lobes [17]. These data suggest endothelial dysfunction in PVS, but mainly in the HP/LF territory. To further test whether this endothelial dysfunction comprised a change in NO, we also investigated bradykinin-induced vasodilation after administration of the eNOS inhibitor L-NAME. However, L-NAME attenuated the bradykinin-induced vasodilation to a similar extent in pulmonary small arteries from all territories, suggesting that the decreased vasodilation to bradykinin was NO-independent. This is consistent with unchanged eNOS protein expression in the HP/LF lobes compared to sham. Bradykinin also exerts a vasodilator effect via endothelium-derived hyperpolarizing factor and via prostacyclin. The decreased vasodilation to bradykinin in the PVB HP/LF territory was thus the result of either decreased endothelium-derived hyperpolarizing factor or decreased prostacyclin. It could therefore be speculated that prostacyclin therapy would mainly target the HP/HF vasculature, and could be the therapy of choice for PVS-induced PH. As we did not directly test the effect of prostacyclin therapy in vivo or in vitro, future studies are needed to assess this possibility.

This study was the first to test the effect of two first-line therapies, i.e., the NO donor SNP and PDE5 inhibitor sildenafil, separately in the unaffected HP/HF and the affected HP/LF vasculatures. The vasodilator effect of the NO donor was not different in the HP/HF PVB upper vs. the HP/LF PVB lower lobes. There was, however, a difference in efficacy of PDE5 inhibition by sildenafil. Unfortunately, the effect was bigger in the HP/LF vasculature. Although this is theoretically unfavourable, we did not find any evidence for pulmonary oedema in vivo (no reduction in arterial pO_2_, Figure 3). The reduced effect of sildenafil in vitro could potentially be explained by the lack of flow in the in vitro experiments. It is possible that in vivo, in the presence of flow, shear-stress-induced NO production is higher in the HP/HF territories as compared to the HP/LF territories, and, consequently, that an increase in cGMP in response to PDE5 inhibition may be more pronounced in the HP/HF territories. In group I PH, vascular narrowing is also believed to be heterogeneous. Rol et al. found that in idiopathic PAH patients, only 30% of the vessels have a reduced inner diameter [27]. These vessels will be HP/LF, while the remaining 70% of the vasculature will be HP/HF. The difference from PVS-induced PH is that in group I PH, the HP/LF vasculature should be targeted. Extrapolation of our results could possibly explain the efficacy of PDE5 inhibition in PAH.

### 4.2. Clinical Relevance

Both surgical and catheter-based treatments of PVS in the paediatric population are challenging, and restenosis is frequent. However, it is currently the only option to prolong survival for severe progressive PVS. Pulmonary vasodilators could reduce RV afterload and postpone the need for surgery, but may induce pulmonary oedema, particularly when vasodilation occurs in both stenotic and unaffected territories. Currently, approximately 30% of paediatric patients with PVS are on pulmonary vasodilator treatment, with sildenafil being the most commonly administered drug [7,24]. Our study showed that, although the pulmonary small arteries from the HP/LF territories respond better to sildenafil in vitro than those from HP/HF territories, sildenafil treatment did not result in overt pulmonary oedema in vivo. Currently, there have been no studies comparing the effects of different vasoactive drugs as treatment for PVS-induced PH. We previously showed that ET_A/B_ blockade with tezosentan also induced pulmonary vasodilation. However, the pulmonary vasodilator effect of sildenafil in the present study was approximately twice as big as the pulmonary vasodilation induced by tezosentan. Similarly, the effect of sildenafil was larger than that of the NO donor SNP. Hence, sildenafil should be the treatment of choice for PVS-induced PH.

## Figures and Tables

**Figure 1 jcm-08-01204-f001:**
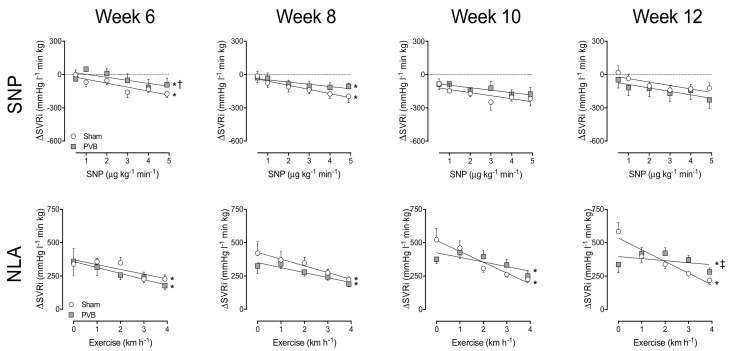
Systemic vasodilation by different doses of the nitric oxide donor sodium nitroprusside (SNP, upper panel), and the endothelial nitric oxide synthase inhibitor N^ω^-nitro-L-arginine (NLA) at rest and during graded treadmill exercise (lower panels) at 6, 8, 10 and 12 weeks after pulmonary vein banding (PVB) or sham surgery. ΔSVRi: Change in systemic vascular resistance index; * *p* ≤ 0.05 effect exercise; † *p* ≤ 0.05 vs. sham; ‡ *p* ≤ 0.05 effect exercise PVB vs. effect exercise sham by two-way ANOVA. Values are means ± SEM.

**Figure 2 jcm-08-01204-f002:**
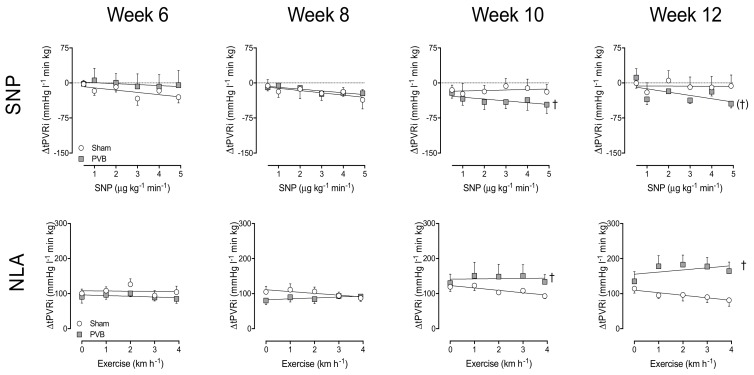
Pulmonary vasodilation by different doses of the nitric oxide donor sodium nitroprusside (SNP, upper panel), and the endothelial nitric oxide synthase inhibitor N^ω^-nitro-L-arginine (NLA) at rest and during graded treadmill exercise (lower panels) at 6, 8, 10 and 12 weeks after pulmonary vein banding (PVB) or sham surgery. ΔtPVRi: Change in total pulmonary vascular resistance index; * *p* ≤ 0.05 effect exercise; † *p* ≤ 0.05, (†) *p* ≤ 0.10 vs. sham. Values are means ± SEM.

**Figure 3 jcm-08-01204-f003:**
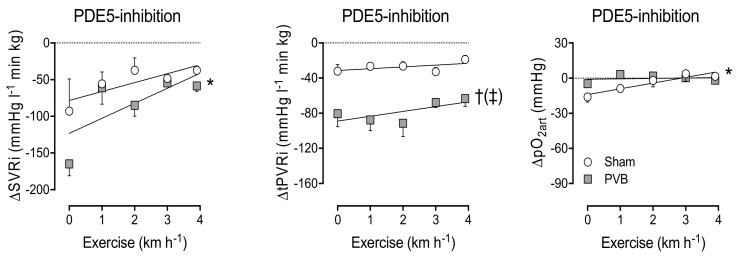
Changes in systemic vascular resistance index (ΔSVRi, left panel), total pulmonary vascular resistance index (ΔtPVRi, middle panel) and arterial partial oxygen pressure (ΔpO_2art_, right panel), induced by vasodilation to phosphodiesterase-5 (PDE5) inhibitor sildenafil, at rest and during graded treadmill exercise. * *p* ≤ 0.05 effect exercise; † *p* ≤ 0.05 vs. sham; (‡) *p* ≤ 0.10 effect exercise PVB different from effect exercise sham by two-way ANOVA. Values are means ± SEM.

**Figure 4 jcm-08-01204-f004:**
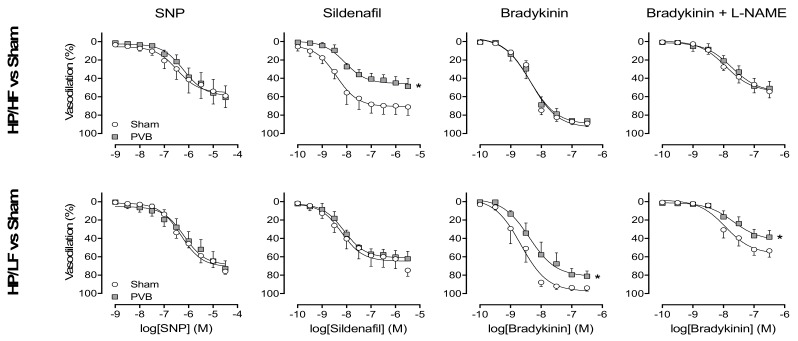
Small pulmonary arteries of the high pressure/high flow (HP/HF) and high pressure/low flow (HP/LF) territories of pulmonary vein banding (PVB) swine, or small pulmonary arteries from corresponding lobes of sham animals, were isolated, pre-constricted by u46619 and exposed to incremental doses of the nitric oxide donor sodium nitroprusside (SNP), phosphodiesterase-5 (PDE5) inhibitor sildenafil or bradykinin with or without prior inhibition of endothelial nitric oxide synthase by L-N^ω^-nitroarginine methyl ester (L-NAME). Data are presented as percentages of preconstriction with U46619. * *p* ≤ 0.05 vs. corresponding sham, by regression analyses. Values are means ± SEM.

**Figure 5 jcm-08-01204-f005:**
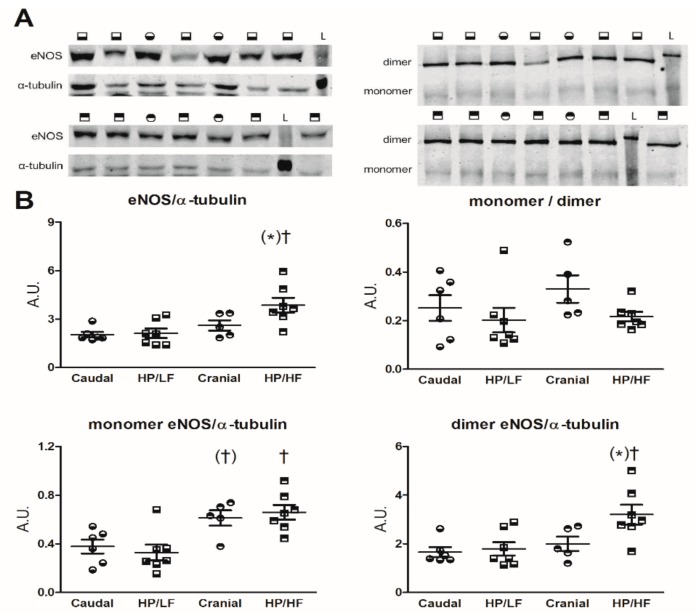
Endothelial nitric oxide syntase eNOS protein expression in lung tissue of the high pressure/high flow (HP/HF) and high pressure/low flow (HP/LF) territories of pulmonary vein banding (PVB) swine, and corresponding caudal and cranial lobes from sham-operated swine. (**A**) typical examples of original blots, with symbols correponding to (**B**), denoting the different lung territories and L denoting the ladder. (**B**) Detection of eNOS monomer and dimer fractions with low temperature sodium dodecyl sulfate–polyacrylamide gel electrophoresis (SDS-PAGE) showed that monomer:dimer ratio was unaltered, while both eNOS monomer and dimer were higher in HP/HF as compared to HP/LF, but the eNOS dimer only tended to be higher in HP/HF vs. corresponding sham. Data are presented as arbitrary units (A.U.) defined as the ratio of intensity of the eNOS band and the α-tubulin band or the eNOS monomer and dimer bands. Data are shown as dotplots. Lines denote means ± SEM. (*) *p* ≤ 0.10 vs. corresponding sham (caudal or cranial), † *p* ≤ 0.05, (†) *p* ≤ 0.10 HP/HF vs. HP/LF or cranial vs. caudal by two-way ANOVA.

**Table 1 jcm-08-01204-t001:** Haemodynamics at rest and during exercise over the development of post capillary pulmonary hypertension (PH).

				Week															
	*n*	Group			6				8				10				12		
MAP	5	Sham	Rest	90	±	1		81	±	4		92	±	5		95	±	4	
(mmHg)	5		Exercise	93	±	4		92	±	5		98	±	4		102	±	3	
	6	PVB	Rest	86	±	5		85	±	4		80	±	4		86	±	3	(†)
	6		Exercise	90	±	2		88	±	3		83	±	4	†	86	±	7	(†)
LAP	5	Sham	Rest	6.0	±	2.1		3.8	±	1.2		4.0	±	1.2		3.1	±	1.0	
(mmHg)	5		Exercise	9.5	±	1.2		9.0	±	0.8	*	7.9	±	0.9	(*)	10.1	±	1.1	*
	6	PVB	Rest	4.1	±	0.9		3.5	±	1.3		6.4	±	1.3		7.1	±	1.6	
	6		Exercise	7.2	±	0.9	*	5.9	±	0.8	†	10.1	±	1.7		12.8	±	1.9	* (‡)
mPAP	5	Sham	Rest	20	±	1		17	±	2		22	±	1		20	±	1	
(mmHg)	5		Exercise	34	±	1	*	31	±	1	*	32	±	2	*	35	±	3	*
	6	PVB	Rest	30	±	2	†	31	±	2	†	36	±	3	†	39	±	2	† ‡
	6		Exercise	51	±	2	* †	56	±	3	* †	56	±	2	* †	62	±	3	* † ‡
HR	5	Sham	Rest	152	±	12		136	±	10		138	±	12		132	±	9	
(bpm)	5		Exercise	259	±	10	*	228	±	15	*	243	±	19	*	251	±	10	*
	5	PVB	Rest	147	±	8		139	±	4		136	±	10		135	±	5	
	5		Exercise	246	±	11	*	238	±	11	*	206	±	8	* (†) ‡	207	±	6	* † ‡
CI	5	Sham	Rest	0.19	±	0.01		0.19	±	0.01		0.19	±	0.01		0.17	±	0.01	
(L min^−1^ kg^−1^)	5		Exercise	0.32	±	0.01	*	0.30	±	0.01	*	0.29	±	0.02	*	0.28	±	0.02	*
	5	PVB	Rest	0.21	±	0.01		0.19	±	0.01		0.18	±	0.02		0.16	±	0.01	(‡)
	5		Exercise	0.32	±	0.01	*	0.28	±	0.02	*	0.25	±	0.01	* ‡	0.23	±	0.01	* † ‡
SVi	5	Sham	Rest	1.30	±	0.09		1.39	±	0.09		1.41	±	0.13		1.31	±	0.11	
(mL kg^−1^)	5		Exercise	1.26	±	0.06		1.36	±	0.11		1.23	±	0.14		1.11	±	0.10	
	5	PVB	Rest	1.41	±	0.09		1.35	±	0.09		1.38	±	0.05		1.21	±	0.06	
	5		Exercise	1.26	±	0.06		1.19	±	0.05		1.25	±	0.06		1.10	±	0.06	
SVRi	5	Sham	Rest	472	±	18		439	±	28		488	±	24		562	±	26	(‡)
(mmHg L^−1^ min kg)	5		Exercise	288	±	13	*	309	±	22	*	340	±	22	*	378	±	27	* (‡)
	5	PVB	Rest	390	±	32	†	471	±	52		455	±	59		534	±	38	(‡)
	5		Exercise	270	±	13	*	319	±	27	*	325	±	24	(*)	391	±	45	*
tPVRi	5	Sham	Rest	107	±	9		94	±	11		118	±	7		116	±	10	
(mmHg L^−1^ min kg)	5		Exercise	106	±	6		105	±	7		111	±	10		128	±	11	
	5	PVB	Rest	132	±	4	†	169	±	14	†	192	±	12	† ‡	244	±	26	† ‡
	5		Exercise	155	±	7	* †	201	±	16	† (‡)	218	±	11	† ‡	274	±	14	† ‡
pO_2art_	5	Sham	Rest	98	±	4		102	±	4		103	±	4		102	±	4	
(mmHg)	5		Exercise	84	±	4	*	86	±	3	*	87	±	3	*	92	±	5	
	6	PVB	Rest	92	±	5		87	±	5	†	88	±	3	†	88	±	3	†
	6		Exercise	68	±	3	* †	63	±	2	* †	66	±	3	* †	65	±	3	* †

Haemodynamics obtained 6, 8, 10 and 12 weeks after banding at rest and during exercise (4 km h^−1^). MAP: mean arterial pressure, LAP: mean left atrial pressure, mPAP: mean pulmonary arterial pressure, HR: heart rate, CI: cardiac index, SVi: stroke volume index (CI/HR), SVRi: systemic vascular resistance index (MAP/CI), tPVRi: total pulmonary vascular resistance index (mPAP/CI), pO_2art_: arterial partial oxygen pressure. Data are means ± SEM. * *p* ≤ 0.05, (*) *p* ≤ 0.1 vs. rest, † *p* ≤ 0.05, (†) *p* ≤ 0.1 vs. sham. ‡ *p* ≤ 0.05, (‡) *p* ≤ 0.1 vs. week 6.

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
