# Peer review of "Intervening with the Nitric Oxide Pathway to Alleviate Pulmonary Hypertension in Pulmonary Vein Stenosis"

_jcm, 2019, doi:10.3390/jcm8081204_

Round 1

Reviewer 1 Report

The study led by Van Duin et al., has critically analyzed the outcomes of vasodilator therapies in high pressure low flow and high pressure high flow regions on the lung using a newly developed Swine model. The present study has tested the effectiveness of different vasoactive drugs for the pulmonary vein stenosis induced pulmonary hypertension and recommends the targeting of high pressure high flow regions of the vein. This work may add value to the understanding of PVS induced PH and lead to better treatment options. However, authors need to address the following concerns in the manuscript.

Authors mention that there is no alteration in the systemic circulation parameters however LAP (8 weeks); HR (12 weeks); MAP (10 weeks) show significant change compared to Sham. Authors can explain the difference between (†) and † symbols in Table 1. If they are different then the significance value for (†) is missing in the Legend for Table 1. Figure 2 legends mentions ‡ this symbol but it is missing on the figure. Figure 5 legends, authors should expand what is A.U mentioned on the y axis of the figures. Also, line 321 mentions HF/LF instead HP/LF. Line 30, in the abstract HP/LF is repeated.

Author Response

We would like to thank the reviewer for his/her kind comments and critical reading of the manuscript. A point to point response is given below. 

Authors mention that there is no alteration in the systemic circulation parameters however LAP (8 weeks); HR (12 weeks); MAP (10 weeks) show significant change compared to Sham.

We have reworded the first two paragraphs of the results section and now specifically mentioned systemic hemodynamic variables that were different either at rest or during exercise (lines 582-596).

Authors can explain the difference between (†) and † symbols in Table 1. If they are different then the significance value for (†) is missing in the Legend for Table 1.

We apologize for the oversight. Symbols between () present a P-value <0.1, which has now been stated in the legend (Line 725).

Figure 2 legends mentions ‡ this symbol but it is missing on the figure.

We re-checked the statistical analysis and there is indeed no difference in the exercise response between sham and PVB. We have removed the symbol from the legend of figure 2.

Figure 5 legends, authors should expand what is A.U mentioned on the y axis of the figures.

Data in figure 5 are presented as arbitrary units (A.U.) defined as the ratio of intensity of the eNOS band and the α-tubulin band or the eNOS monomer and dimer bands. This has no been stated in the legend of Figure 5 (lines 812-813) as well as in the Methods section (Lines 574-578).

Also, line 321 mentions HF/LF instead HP/LF. Line 30, in the abstract HP/LF is repeated.

Thank you for pointing out these errors,they have now been corrected (line 36)

Reviewer 2 Report

Dear editors, The manuscript entitled “Intervening with the Nitric Oxide Pathway to Alleviate Pulmonary Hypertension in Pulmonary Vein Stenosis” by Richard W. B. van Duin et.al presents some interesting experimental data. The authors examined two first-line vasodilator therapies of Pulmonary Hypertension (ie PDE5-inhibitor Sildenafil and nitric oxide) in pulmonary vein banding in Swine model with stenosis causing increase in PVR and mPAP over time. Pulmonary vein banding model allowed investigating into two subsets of arteries from the HP/HF and HP/LF-territories and sensitivity of isolated pulmonary small arteries to vasodilators NO-donor SNP, PDE5-inhibitor Sildenafil are also checked. Moreover, exercise testing also allowed the authors to detect the perturbations in cardiopulmonary function that may not be apparent under resting conditions. Besides, discussion and methods were well written and cohesive. It is recommended the authors address comments/concerns listed below.

comments:

*In figure 5 eNOS monomer and dimer are higher in high pressure / high flow (HP/HF) as compared to high pressure / low flow (HP/LF) in lung tissue territories of pulmonary vein banding (PVB) swine. It would be great if the accompanying western blots were also depicted in the figure.

* Results like, influence of exercise on the endogenous NO-mediated vasodilator in the pulmonary vasculature of PVB and its effects over time or missing in the abstract

Author Response

We would like to thank the reviewer for his/her positive remarks. A point to point reply to the comments is provided below.

*In figure 5 eNOS monomer and dimer are higher in high pressure / high flow (HP/HF) as compared to high pressure / low flow (HP/LF) in lung tissue territories of pulmonary vein banding (PVB) swine. It would be great if the accompanying western blots were also depicted in the figure.

We have now added typical examples of the Western blots to Figure 5. The symbols above the blots reflect the different lung territories and correspond to the symbols in the dot-plots.

*Results like, influence of exercise on the endogenous NO-mediated vasodilator in the pulmonary vasculature of PVB and its effects over time or missing in the abstract

We have modified the abstract according to the reviewers suggestions.